# On the training dynamics of deep networks with $L_2$ regularization

**Aitor Lewkowycz**
Google
Mountain View, CA
alewkowycz@google.com

**Guy Gur-Ari**
Google
Mountain View, CA
guyga@google.com

## Abstract

We study the role of $L_2$ regularization in deep learning, and uncover simple relations between the performance of the model, the $L_2$ coefficient, the learning rate, and the number of training steps. These empirical relations hold when the network is overparameterized. They can be used to predict the optimal regularization parameter of a given model. In addition, based on these observations we propose a dynamical schedule for the regularization parameter that improves performance and speeds up training. We test these proposals in modern image classification settings. Finally, we show that these empirical relations can be understood theoretically in the context of infinitely wide networks. We derive the gradient flow dynamics of such networks, and compare the role of $L_2$ regularization in this context with that of linear models.

## 1 Introduction

Machine learning models are commonly trained with $L_2$ regularization. This involves adding the term $\frac{1}{2}\lambda\|\theta\|_2^2$ to the loss function, where $\theta$ is the vector of model parameters and $\lambda$ is a hyperparameter. In some cases, the theoretical motivation for using this type of regularization is clear. For example, in the context of linear regression, $L_2$ regularization increases the bias of the learned parameters while reducing their variance across instantiations of the training data; in other words, it is a manifestation of the bias-variance tradeoff. In statistical learning theory, a "hard" variant of $L_2$ regularization, in which one imposes the constraint $\|\theta\|_2 \leq \epsilon$, is often employed when deriving generalization bounds.

In deep learning, the use of $L_2$ regularization is prevalent and often leads to improved performance in practical settings [Hinton, 1986], although the theoretical motivation for its use is less clear. Indeed, it well known that overparameterized models overfit far less than one may expect [Zhang et al., 2016], and so the classical bias-variance tradeoff picture does not apply [Neyshabur et al., 2017, Belkin et al., 2018, Geiger et al., 2020]. There is growing understanding that this is caused, at least in part, by the (implicit) regularization properties of stochastic gradient descent (SGD) [Soudry et al., 2017]. The goal of this paper is to improve our understanding of the role of $L_2$ regularization in deep learning.

### 1.1 Our contribution

We study the role of $L_2$ regularization when training over-parameterized deep networks, taken here to mean networks that can achieve training accuracy 1 when trained with SGD. Specifically, we consider the early stopping performance of a model, namely the maximum test accuracy a model achieves during training, as a function of the $L_2$ parameter $\lambda$. We make the following observations based on the experimental results presented in the paper.

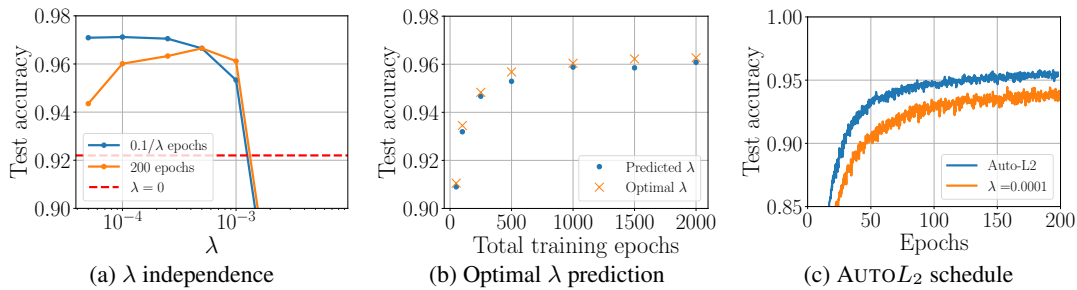

(a) $\lambda$ independence      (b) Optimal $\lambda$ prediction      (c) AUTO$L_2$ schedule

Figure 1: Wide ResNet 28-10 trained on CIFAR-10 with momentum and data augmentation. (a) Final test accuracy vs. the $L_2$ parameter $\lambda$. When the network is trained for a fixed amount of epochs, optimal performance is achieved at a certain value of $\lambda$. But when trained for a time proportional to $\lambda^{-1}$, performance plateaus and remains constant down to the lowest values of $\lambda$ tested. This experiment includes a learning rate schedule. (b) Test accuracy vs. training epochs for predicted optimal $L_2$ parameter compared with the tuned parameter. (c) Training curves with our dynamical $L_2$ schedule, compared with a tuned, constant $L_2$ parameter.

1. The number of SGD steps until a model achieves maximum performance is $t_* \approx \frac{c}{\lambda}$, where $c$ is a coefficient that depends on the data, the architecture, and all other hyperparameters. We find that this relationship holds across a wide range of $\lambda$ values.

2. If we train with a fixed number of steps, model performance peaks at a certain value of the $L_2$ parameter. However, if we train for a number of steps proportional to $\lambda^{-1}$ then performance improves with decreasing $\lambda$. In such a setup, performance becomes independent of $\lambda$ for sufficiently small $\lambda$. Furthermore, performance with a small, non-zero $\lambda$ is often better than performance without any $L_2$ regularization.

Figure 1a shows the performance of an overparameterized network as a function of the $L_2$ parameter $\lambda$. When the model is trained with a fixed steps budget, performance is maximized at one value of $\lambda$. However, when the training time is proportional to $\lambda^{-1}$, performance improves and approaches a constant value as we decrease $\lambda$.

As we demonstrate in the experimental section, these observations hold for a variety of training setups which include different architectures, data sets, and optimization algorithms. In particular, when training with vanilla SGD (without momentum), we observe that the number of steps until maximum performance depends on the learning rate $\eta$ and on $\lambda$ as $t_* \approx \frac{c'}{\eta \cdot \lambda}$. The performance achieved after this many steps depends only weakly on the choice of learning rate.

**Applications.** We present two practical applications of these observations. First, we propose a simple way to predict the optimal value of the $L_2$ parameter, based on a cheap measurement of the coefficient $c$. Figure 1b compares the performance of models trained with our predicted $L_2$ parameter with that of models trained with a tuned parameter. In this realistic setting, we find that our predicted parameter leads to performance that is within 0.4% of the tuned performance on CIFAR-10, at a cost that is marginally higher than a single training run. As shown below, we also find that the predicted parameter is consistently within an order of magnitude of the optimal, tuned value.

As a second application we propose AUTO$L_2$, a dynamical schedule for the $L_2$ parameter. The idea is that large $L_2$ values achieve worse performance but also lead to faster training. Therefore, in order to speed up training one can start with a large $L_2$ value and decay it during training (this is similar to the intuition behind learning rate schedules). In Figure 1c we compare the performance of a model trained with AUTO$L_2$ against that of a tuned but constant $L_2$ parameter, and find that AUTO$L_2$ outperforms the tuned model both in speed and in performance.

**Learning rate schedules.** Our empirical observations apply in the presence of learning rate schedules. In particular, Figure 1a shows that the test accuracy remains approximately the same if we scale the training time as $1/\lambda$. As to our applications, in section 3 we propose an algorithm for predicting the optimal $L_2$ value in the presence of learning rate schedules, and the predicted value

gives comparable performance to the tuned result. As to the AUTO$L_2$ algorithm, we find that in the presence of learning rate schedules it does not perform as well as a tuned but constant $L_2$ parameter. We leave combining AUTO$L_2$ with learning rate schedules to future work.

**Theoretical contribution.** Finally, we turn to a theoretical investigation of the empirical observations made above. As a first attempt at explaining these effects, consider the following argument based on the loss landscape. For overparameterized networks, the Hessian spectrum evolves rapidly during training [Sagun et al., 2017, Gur-Ari et al., 2018, Ghorbani et al., 2019]. After a small number of training steps with no $L_2$ regularization, the minimum eigenvalue is found to be close to zero. In the presence of a small $L_2$ term, we therefore expect that the minimal eigenvalue will be approximately $\lambda$. In quadratic optimization, the convergence time is inversely proportional to the smallest eigenvalue of the Hessian.[1] Based on this intuition, we may then expect that convergence time will be proportional to $\lambda^{-1}$. The fact that performance is roughly constant for sufficiently small $\lambda$ can then be explained if overfitting can be mostly attributed to optimization in the very low curvature directions [Rahaman et al., 2018]. Now, our empirical finding is that the time it takes the network to reach maximum accuracy is proportional to $\lambda^{-1}$. In some cases this is the same as the convergence time, but in other cases (see for example Figure 4a) we find that performance decays after peaking and so convergence happens later. Therefore, the loss landscape-based explanation above is not sufficient to fully explain the effect.

To gain a better theoretical understanding, we consider the setup of an infinitely wide neural network trained using gradient flow. We focus on networks with positive-homogeneous activations, which include deep networks with ReLU activations, fully-connected or convolutional layers, and other common components. By analyzing the gradient flow update equations of such networks, we are able to show that the performance peaks at a time of order $\lambda^{-1}$ and deteriorates thereafter. This is in contrast to the performance of linear models with $L_2$ regularization, where no such peak is evident. These results are consistent with our empirical observations, and may help shed light on the underlying causes of these effects.

According to known infinite width theory, in the absence of explicit regularization, the kernel that controls network training is constant [Jacot et al., 2018]. Our analysis extends the known results on infinitely wide network optimization, and indicates that the kernel decays in a predictable way in the presence of $L_2$ regularization. We hope that this analysis will shed further light on the observed performance gap between infinitely wide networks which are under good theoretical control, and the networks trained in practical settings [Arora et al., 2019, Novak et al., 2019, Wei et al., 2018, Lewkowycz et al., 2020].

**Related works.** $L_2$ regularization in the presence of batch-normalization [Ioffe and Szegedy, 2015] has been studied in [van Laarhoven, 2017, Hoffer et al., 2018, Zhang et al., 2018]. These papers discussed how the effect of $L_2$ on scale invariant models is merely of having an effective learning rate (and no $L_2$). This was made precise in Li and Arora [2019] where they showed that this effective learning rate is $\eta_{\text{eff}} = \eta e^{2\eta\lambda t}$ (at small learning rates). Our theoretical analysis of large width networks will have has the same behaviour when the network is scale invariant. Finally, in parallel to this work, Li et al. [2020] carried out a complementary analysis of the role of $L_2$ regularization in deep learning using a stochastic differential equation analysis. Their conclusions regarding the effective learning rate in the presence of $L_2$ regularization are consistent with our observations.

## 2  Experiments

**Performance and time scales.** We now turn to an empirical study of networks trained with $L_2$ regularization. In this section we present results for a fully-connected network trained on MNIST, a Wide ResNet [Zagoruyko and Komodakis, 2016] trained on CIFAR-10, and CNNs trained on CIFAR-10. The experimental details are in SM A. The empirical findings discussed in section 1.1 hold across this variety of overparameterized setups.

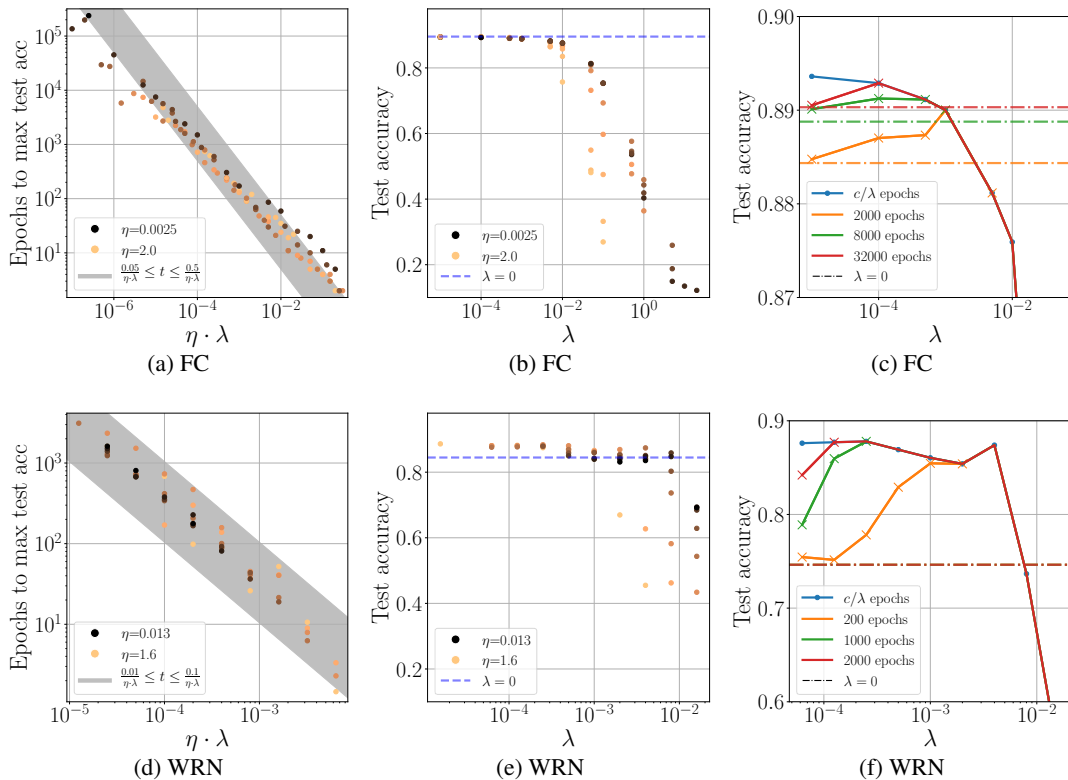

Figure 2: Sweep over $\eta$ and $\lambda$ illustrating how smaller $\lambda$'s require longer times to achieve the same performance. In the left, middle plots, the learning rates are logarithmically spaced between the values displayed in the legend, the specific values are in the SM A. **Left:** Epochs to maximum test accuracy (within .5%), **Middle:** Maximum test accuracy (the $\lambda = 0$ line denotes the maximum test accuracy achieved among all learning rates), **Right:** Maximum test accuracy for a fixed time budget. **(a,b,c)** Fully connected 3-hidden layer neural network evaluated in 512 MNIST samples, evolved for $t \cdot \eta \cdot \lambda = 2$. $\eta = 0.15$ in (c). **(d,e,f)** A Wide Residual Network 28-10 trained on CIFAR-10 without data augmentation, evolved for $t \cdot \eta \cdot \lambda = 0.1$. In (f), $\eta = 0.2$. The $\lambda = 0$ line was evolved for longer than the smallest $L_2$ but there is still a gap.

Figure 2 presents experimental results on fully-connected and Wide ResNet networks. Figure 3 presents experiments conducted on CNNs. We find that the number of steps until optimal performance is achieved (defined here as the minimum time required to be within .5% of the maximum test accuracy) scales as $\lambda^{-1}$, as discussed in Section 1.1. Our experiments span 6 decades of $\eta \cdot \lambda$ (larger $\eta, \lambda$ won't train at all and smaller would take too long to train). Moreover, when we evolved the networks until they have reached optimal performance, the maximum test accuracy for smaller $L_2$ parameters did not get worse. We compare this against the performance of a model trained with a fixed number of epochs, reporting the maximum performance achieved during training. In this case, we find that reducing $\lambda$ beyond a certain value does hurt performance.

While here we consider the simplified set up of vanilla SGD and no data augmentation, our observations also hold in the presence of momentum and data augmentation, see SM C.2 for more experiments. We would like to emphasize again that while the smaller $L_2$ models can reach the same test accuracy as its larger counterparts, models like WRN28-10 on CIFAR-10 need to be trained for a considerably larger number of epochs to achieve this.[2]

**Learning rate schedules.** So far we considered training setups that do not include learning rate schedules. Figure 1a shows the results of training a Wide ResNet on CIFAR-10 with a learning rate schedule, momentum, and data augmentation. The schedule was determined as follows. Given a total

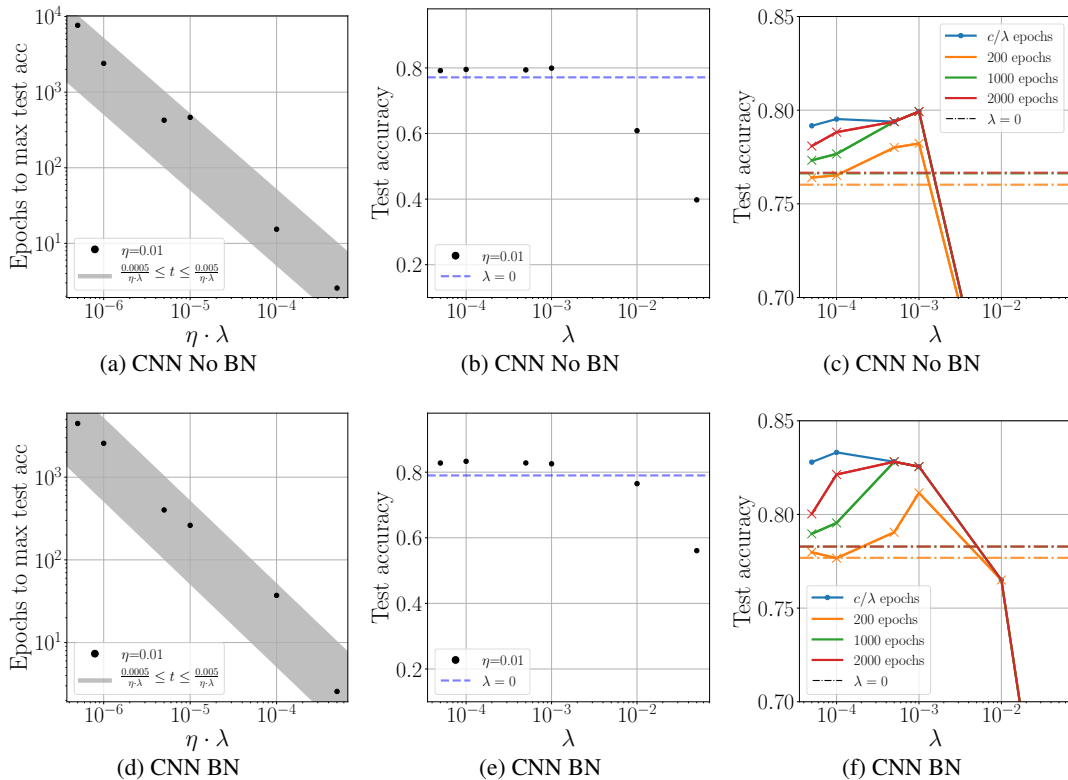

Figure 3: CNNs trained with and without batch-norm with learning rate $\eta = 0.01$. Presented results follow the same format as Figure 2.

number of epochs $T$, the learning rate is decayed by a factor of $0.2$ at epochs $\{0.3 \cdot T, 0.6 \cdot T, 0.9 \cdot T\}$. We compare training with a fixed $T$ against training with $T \propto \lambda^{-1}$. We find that training with a fixed budget leads to an optimal value of $\lambda$, below which performance degrades. On the other hand, training with $T \propto \lambda^{-1}$ leads to improved performance at smaller $\lambda$, consistent with our previous observations.

## 3   Applications

We now discuss two practical applications of the empirical observations made in the previous section.

**Optimal $L_2$.** We observed that the time $t_*$ to reach maximum test accuracy is proportional to $\lambda^{-1}$, which we can express as $t_* \approx \frac{c}{\lambda}$. This relationship continues to hold empirically even for large values of $\lambda$. When $\lambda$ is large, the network attains its (significantly degraded) maximum performance after a relatively short amount of training time. We can therefore measure the value of $c$ by training the network with a large $L_2$ parameter until its performance peaks, at a fraction of the cost of a normal training run.

Based on our empirical observations, given a training budget $T$ we predict that the optimal $L_2$ parameter can be approximated by $\lambda_{\text{pred}} = c/T$. This is the smallest $L_2$ parameter such that model performance will peak within training time $T$. Figure 1b shows the result of testing this prediction in a realistic setting: a Wide ResNet trained on CIFAR-10 with momentum$= 0.9$ , learning rate $\eta = 0.2$ and data augmentation. The model is first trained with a large $L_2$ parameter for 2 epochs in order to measure $c$, and we find $c \approx 0.0066$, see figure 4a. We then compare the tuned value of $\lambda$ against our prediction for training budgets spanning close to two orders of magnitude, and find excellent agreement: the predicted $\lambda$'s have a performance which is rather close to the optimal one. Furthermore, the tuned values are always within an order of magnitude of our predictions see figure 4b.

So far we assumed a constant learning rate. In the presence of learning rate schedules, one needs to adjust the prediction algorithm. Here we address this for the case of a piecewise-constant schedule. For compute efficiency reasons, we expect that it is beneficial to train with a large learning rate as long as accuracy continues to improve, and to decay the learning rate when accuracy peaks. Therefore, given a fixed learning rate schedule, we expect the optimal $L_2$ parameter to be the one at which accuracy peaks at the time of the first learning rate decay. Our prediction for the optimal parameter is then $\lambda_{\mathrm{pred}} = c/T_1$, where $T_1$ is the time of first learning rate decay, and the coefficient $c$ is measured as before with a fixed learning rate. In our experiments, this prediction is consistently within an order of magnitude of the optimal parameter, and gives comparable performance. For example, in the case of Figure 1a with $T = 200$ epochs and $T_1 = 0.3T$, we find $\lambda_{\mathrm{pred}} \approx 0.0001$ (leading to test accuracy 0.960), compared with the optimal value 0.0005 (with test accuracy 0.967).

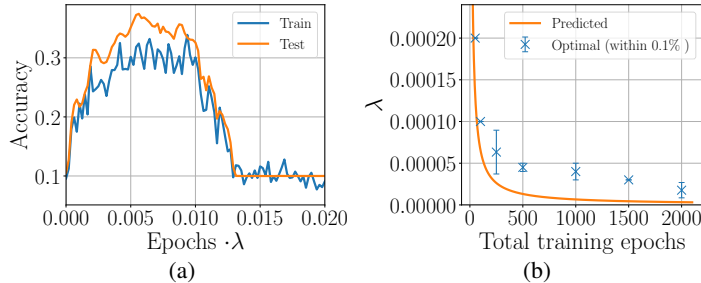

Figure 4: Wide ResNet trained with momentum and data augmentation. (a) We train the model with a large $L_2$ parameter $\lambda = 0.01$ for 2 epochs and measure the coefficient $c = t_* \cdot \lambda \approx 0.0066$, representing the approximate point along the $x$ axis where accuracy is maximized. (b) Optimal (tuned) $\lambda$ values compared with the theoretical prediction. The error bars represent the spread of values that achieve within $0.1\%$ of the optimal test accuracy.

$\textsc{Auto}L_2$: **Automatic $L_2$ schedules.** We now turn to another application, based on the observation that models trained with larger $L_2$ parameters reach their peak performance faster. It is therefore plausible that one can speed up the training process by starting with a large $L_2$ parameter, and decaying it according to some schedule. Here we propose to choose the schedule dynamically by decaying the $L_2$ parameter when performance begins to deteriorate. See SM E for further details.

$\textsc{Auto}L_2$ is a straightforward implementation of this idea: We begin training with a large parameter, $\lambda = 0.1$, and we decay it by a factor of 10 if either the empirical loss (the training loss without the $L_2$ term) or the training error increases. To improve stability, immediately after decaying we impose a refractory period during which the parameter cannot decay again. Figure 1c compares this algorithm against the model with the optimal $L_2$ parameter. We find that $\textsc{Auto}L_2$ trains significantly faster and achieves superior performance. See SM E for other architectures.

In other experiments we have found that this algorithm does not yield improved results when the training procedure includes a learning rate schedule. We leave the attempt to effectively combine learning rate schedules with $L_2$ schedules to future work.

## 4 Theoretical results

We now turn to a theoretical analysis of the training trajectory of networks trained with $L_2$ regularization. We focus on infinitely wide networks with positively-homogeneous activations. Consider a network function $f : \mathbb{R}^d \to \mathbb{R}$ with model parameter $\theta \in \mathbb{R}^p$. The network initialized using NTK parameterization [Jacot et al., 2018]: the initial parameters are sampled i.i.d. from $\mathcal{N}(0, 1)$. The model parameters are trained using gradient flow with loss $L_{\mathrm{tot}} = L + \frac{\lambda}{2}\|\theta\|_2^2$, where $L = \sum_{(x,y)\in S} \ell(x, y)$ is the empirical loss, $\ell$ is the sample loss, and $S$ is the training set of size $N_{\mathrm{samp}}$.

We say that the network function is $k$-homogeneous if $f_{\alpha\theta}(x) = \alpha^k f_\theta(x)$ for any $\alpha > 0$. As an example, a fully-connected network with $L$ layers and ReLU or linear activations is $L$-homogeneous. Networks made out of convolutional, max-pooling or batch-normalization layers are

also $k$-homogeneous.[3] See Li and Arora [2019] for a discussion of networks with homogeneous activations.

Jacot et al. [2018] showed that when an infinitely wide, fully-connected network is trained using gradient flow (and without $L_2$ regularization), its network function obeys the differential equation $\frac{df}{dt}(x) = -\sum_{x' \in S} \Theta_0(x, x') \ell'(x')$, where $t$ is the gradient flow time and $\Theta_t(x, x') = \nabla_\theta f_t(x)^T \nabla_\theta f_t(x)$ is the Neural Tangent Kernel (NTK).

Dyer and Gur-Ari [2020] presented a conjecture that allows one to derive the large width asymptotic behavior of the network function, the Neural Tangent Kernel, as well as of combinations involving higher-order derivatives of the network function. In what follows, we will assume the validity of this conjecture. The following is our main theoretical result.

**Theorem 1.** *Consider a $k$-homogeneous network, and assume that the network obeys the correlation function conjecture of Dyer and Gur-Ari [2020]. In the infinite width limit, the network function $f_t(x)$ and the kernel $\Theta_t(x, x')$ evolve according to the following equations at training time $t$.*

$$\frac{df_t(x)}{dt} = -e^{-2(k-1)\lambda t} \sum_{(x',y') \in S} \Theta_0(x, x') \frac{\partial \ell(x', y')}{\partial f_t} - \lambda k f_t(x), \tag{1}$$

$$\frac{d\Theta_t(x, x')}{dt} = -2(k-1)\lambda \Theta_t(x, x'). \tag{2}$$

The proof hinges on the following equation, which holds for $k$-homogeneous functions: $\sum_\mu \theta_\mu \partial_\mu \partial_{\nu_1} \cdots \partial_{\nu_m} f(x) = (k-m) \partial_{\nu_1} \cdots \partial_{\nu_m} f(x)$. This equation allows us to show that the only effect of $L_2$ regularization at infinite width is to introduce simple terms proportional to $\lambda$ in the gradient flow update equations for both the function and the kernel.

We refer the reader to the SM for the proof. We mention in passing that the case $k = 0$ corresponds to a scaling-invariant network function which was studied in Li and Arora [2019]. In this case, training with $L_2$ term is equivalent to training with an exponentially increasing learning rate.

For commonly used loss functions, and for $k > 1$, we expect that the solution obeys $\lim_{t \to \infty} f_t(x) = 0$. We will prove that this holds for MSE loss, but let us first discuss the intuition behind this statement. At late times the exponent in front of the first term in (1) decays to zero, leaving the approximate equation $\frac{df(x)}{dt} \approx -\lambda k f(x)$ and leading to an exponential decay of the function to zero. Both the explicit exponent in the equation, and the approximate late time exponential decay, suggest that this decay occurs at a time $t_{\text{decay}} \propto \lambda^{-1}$. Therefore, we expect that the minimum of the empirical loss to occur at a time proportional to $\lambda^{-1}$, after which the bare loss will increase because the function is decaying to zero. We observe this behaviour empirically for wide fully-connected networks and for Wide ResNet in the SM.

We now focus on MSE loss and solve the gradient flow equation (1) for this case.

**Theorem 2.** *Let the sample loss be $\ell(x, y) = \frac{1}{2}(f(x) - y)^2$, and assume that $k \geq 2$. Suppose that, at initialization, the kernel $\Theta_0$ has eigenvectors $\hat{e}_a \in \mathbb{R}^{N_{\text{samp}}}$ with corresponding eigenvalues $\gamma_a$. Then during gradient flow, the eigenvalues evolve as $\gamma_a(t) = \gamma_a e^{-2(k-1)\lambda t}$ while the eigenvectors are static. Suppose we treat $f \in \mathbb{R}^{N_{\text{samp}}}$ as a vector defined on the training set. Then each mode of the function, $f_a := (\hat{e}_a)^T f \in \mathbb{R}$, evolves independently as*

$$f_a(x; t) = e^{\frac{\gamma_a(t)}{2(k-1)\lambda} - k\lambda t} \left\{ e^{-\frac{\gamma_a}{2(k-1)\lambda}} f_a(x; 0) + \gamma_a y_a \int_0^t dt' \exp \left[ -\frac{\gamma_a(t')}{2(k-1)\lambda} - (k-2)\lambda t' \right] \right\}. \tag{3}$$

*Here, $y_a := (\hat{e}_a)^T y$. At late times, $\lim_{t \to \infty} f_t(x) = 0$ on the training set.*

The properties of the solution (3) depend on whether the ratio $\gamma_a/\lambda$ is greater than or smaller than 1, as illustrated in Figure 5. When $\gamma_a/\lambda > 1$, the function approaches the label mode $y_{\text{mode}} = y_a$ at a time that is of order $1/\gamma_a$. This behavior is the same as that of a linear model, and represents ordinary learning. Later, at a time of order $\lambda^{-1}$ the mode decays to zero as described above; this late time decay is not present in the linear model. Next, when $\gamma_a/\lambda < 1$ the mode decays to zero at a time of order $\lambda^{-1}$, which is the same behavior as that of a linear model.

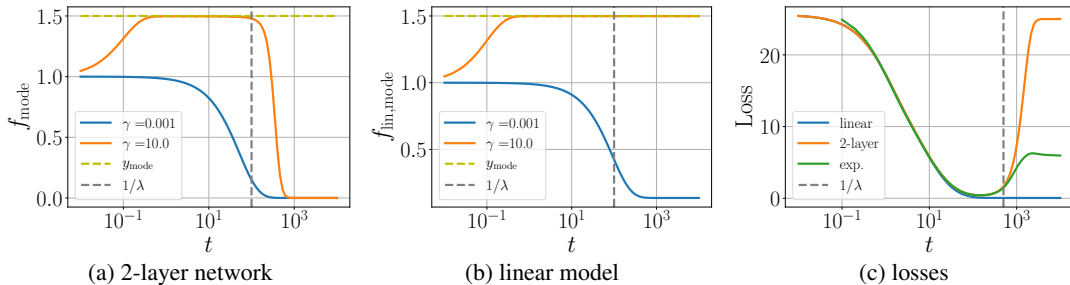

Figure 5: (a) The theoretical evolution of an infinitely wide 2-layer network with $L_2$ regularization ($k = 2$, $\lambda = 0.01$). Two modes are shown, representing small and large ratios $\gamma/\lambda$. (b) The same, for a linear model ($k = 1$). (c) Training loss vs. time for a wide network trained on a subset of MNIST with even/odd labels, with $\lambda = 0.002$. We compare the kernel evolution with gradient descent for a 2-layer ReLU network. The blue and orange curves are the theoretical predictions when setting $k = 1$ and $k = 2$ in the solution (3), respectively. The green curve is the result of a numerical experiment where we train a 2-layer ReLU network with gradient descent. We attribute the difference between the green and orange curves at late times to finite width effects.

**Generalization of wide networks with $L_2$.**   It is interesting to understand how $L_2$ regularization affects the generalization performance of wide networks. This is well understood for the case of linear models, which correspond to $k = 1$ in our notation, to be an instance of the bias-variance tradeoff. In this case, gradient flow converges to the function $f_*(x) = \Theta(x, X)(\Theta + \lambda I)^{-1}(X, X)Y$, where $X \in \mathbb{R}^{N_{\text{samp}} \times d}$ are the training samples, $Y \in \mathbb{R}^{N_{\text{samp}}}$ is are the labels, and $x \in \mathbb{R}^d$ is any input. When $\lambda = 0$, the solution is highly sensitive to small perturbations in the inputs that affect the flat modes of the kernel, because the kernel is inverted in the solution. In other words, the solution has high variance. Choosing $\lambda > 0$ reduces variance by lifting the low kernel eigenvalues and reducing sensitivity on small perturbations, at the cost of biasing the model parameters toward zero.

Let us now return to infinitely wide networks. These behave like linear models with a fixed kernel when $\lambda = 0$, but as we have seen when $\lambda > 0$ the kernel decays exponentially. Nevertheless, we argue that this decay is slow enough such that the training dynamics follow that of the linear model (obtained by setting $k = 1$ in eq. (1)) up until a time of order $\lambda^{-1}$, when the function begins decaying to zero. This can be seen in Figure 5c, which compares the training curves of a linear and a 2-layer network using the same kernel. We see that the agreement extends until the linear model is almost fully trained, at which point the 2-layer model begins deteriorating due to the late time decay. Therefore, if we stop training the 2-layer network at the loss minimum, we end up with a trained and regularized model. It would be interesting to understand how the generalization properties of this model with decaying kernel differ from those of the linear model.

**Finite-width network.**   Theorem 1 holds in the strict large width, fixed $\lambda$ limit for NTK parameterization. At large but finite width we expect (1) to be a good description of the training trajectory at early times, until the kernel and function because small enough such that the finite-width corrections become non-negligible. Our experimental results imply that this approximation remains good until after the minimum in the loss, but that at late times the function will not decay to zero; see for example Figure 5c. See the SM for further discussion for the case of deep linear models. We reserve a more careful study of these finite width effects to future work.

## 5   Discussion

In this work we consider the effect of $L_2$ regularization on overparameterized networks. We make two empirical observations: (1) The time it takes the network to reach peak performance is proportional to $\lambda$, the $L_2$ regularization parameter, and (2) the performance reached in this way is independent of $\lambda$ when $\lambda$ is not too large. We find that these observations hold for a variety of overparameterized training setups; see the SM for some examples where they do not hold. We expect the peak performance to depend on $\lambda$ and $\eta$, but not on other quantities such as the initialization scale. We verify this empirically in SM F.

Motivated by these observations, we suggest two practical applications. The first is a simple method for predicting the optimal $L_2$ parameter at a given training budget. The performance obtained using this prediction is close to that of a tuned $L_2$ parameter, at a fraction of the training cost. The second is AUTO$L_2$, an automatic $L_2$ parameter schedule. In our experiments, this method leads to better performance and faster training when compared against training with a tuned $L_2$ parameter. We find that these proposals work well when training with a constant learning rate; we leave an extension of these methods to networks trained with learning rate schedules to future work.

We attempt to understand the empirical observations by analyzing the training trajectory of infinitely wide networks trained with $L_2$ regularization. We derive the differential equations governing this trajectory, and solve them explicitly for MSE loss. The solution reproduces the observation that the time to peak performance is of order $\lambda^{-1}$. This is due to an effect that is specific to deep networks, and is not present in linear models: during training, the kernel (which is constant for linear models) decays exponentially due to the $L_2$ term.

## Acknowledgments and Disclosure of Funding

The authors would like to thank Yasaman Bahri, Ethan Dyer, Jaehoon Lee, Behnam Neyshabur, and Sam Schoenholz for useful discussions. We especially thank Behnam for encouraging us to use our scaling law observations to come up with a schedule for the $L_2$ parameter.

## Broader Impact

This work does not present any foreseeable societal consequence.

## Footnotes

[1]In linear regression with $L_2$ regularization, optimization is controlled by a linear kernel $K = X^T X + \lambda I$, where $X$ is the sample matrix and $I$ is the identity matrix in parameter space. Optimization in each kernel eigendirection evolves as $e^{-\gamma t}$ where $\gamma$ is the corresponding eigenvalue. When $\lambda > 0$ and the model is overparameterized, the lowest eigenvalue of the kernel will be typically close to $\lambda$, and therefore the time to convergence will be proportional to $\lambda^{-1}$.

[2]The longer experiments ran for 5000 epochs while one usually trains these models for ∼300 epochs.

[3]Batch normalization is often implemented with an $\epsilon$ parameter meant to prevent numerical instabilities. Such networks are only approximately homogeneous.

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
