[Supplementary Material]

# Supplementary material

## A  Experimental details

We are using JAX [Bradbury et al., 2018].

All the models except for section C.4 have been trained with Softmax loss normalized as $\mathcal{L}(\{x,y\}_B) = \frac{1}{2k|B|} \sum_{(x,y) \in B, i} y_i \log p_i(x), p_i(x) = \frac{e^{f^i(x)}}{\sum_j e^{f^j(x)}}$, where $k$ is the number of classes and $y^i$ are one-hot targets.

All experiments that compare different learning rates and $L_2$ parameters use the same seed for the weights at initialization and we consider only one such initialization (unless otherwise stated) although we have not seen much variance in the phenomena described. We will be using standard normalization with LeCun initialization $W \sim \mathcal{N}(0, \frac{\sigma_w^2}{N_{in}}), b \sim \mathcal{N}(0, \sigma_b^2)$.

Batch Norm: we are using JAX's Stax implementation of Batch Norm which doesn't keep track of training batch statistics for test mode evaluation.

Data augmentation: denotes flip, crop and mixup.

We consider 3 different networks:

- WRN: Wide Resnet 28-10 [Zagoruyko and Komodakis, 2016] with has batch-normalization and batch size 1024 (per device batch size of 128), $\sigma_w = 1, \sigma_b = 0$. Trained on CIFAR-10.

- FC: Fully connected, three hidden layers with width 2048 and ReLU activation and batch size 512,$\sigma_w = \sqrt{2}, \sigma_b = 0$. Trained on 512 samples of MNIST.

- CNN: We use the following architecture: $\text{Conv}_1(300) \rightarrow \text{Act} \rightarrow \text{Conv}_2(300) \rightarrow \text{Act} \rightarrow \text{MaxPool}((6,6), \text{'VALID'}) \rightarrow \text{Conv}_1(300) \rightarrow \text{Act} \rightarrow \text{Conv}_2(300) \rightarrow \text{MaxPool}((6,6), \text{'VALID'}) \rightarrow \text{Flatten}() \rightarrow \text{Dense}(500) \rightarrow \text{Dense}(10)$. $\text{Dense}(n)$ denotes a fully-connected layer with output dimension $n$. $\text{Conv}_1(n), \text{Conv}_2(n)$ denote convolutional layers with 'SAME' or 'VALID' padding and $n$ filters, respectively; all convolutional layers use $(3,3)$ filters. $\text{MaxPool}((2,2), \text{'VALID'})$ performs max pooling with 'VALID' padding and a $(2,2)$ window size. Act denotes the activation: '(Batch-Norm $\rightarrow$) ReLU ' depending on whether we use Batch-Normalization or not. We use batch size 128, $\sigma_w = \sqrt{2}, \sigma_b = 0$. Trained on CIFAR-10 without data augmentation.

The WRN experiments are run on v3-8 TPUs and the rest on P100 GPUs.

Here we describe the particularities of each figure. Whenever we report performance for a given time budget, we report the maximum performance during training which does not have to happen at the end of training.

**Figure 1a** WRN trained using momentum$= 0.9$, data augmentation and a learning rate schedule where $\eta(t = 0) = 0.2$ and then decays $\eta \rightarrow 0.2\eta$ at $\{0.3 \cdot T, 0.6 \cdot T, 0.9 \cdot T\}$, where $T$ is the number of epochs. We compare training with a fixed $T = 200$ training budget, against training with $T(\lambda) = 0.1/\lambda$. This was chosen so that $T(0.0005) = 200$.

**Figures 1b, 4, S4.** WRN trained using momentum$= 0.9$, data augmentation and $\eta = 0.2$ for $\lambda \in (5 \cdot 10^{-6}, 10^{-5}, 5 \cdot 10^{-5}, 0.0001, 0.0002, 0.0004, 0.001, 0.002)$. The predicted $\lambda$ performance of 1b was computed at $\lambda = 0.0066/T \in (0.000131, 6.56 \cdot 10^{-5}, 2.63 \cdot 10^{-5}, 1.31 \cdot 10^{-5}, 6.56 \cdot 10^{-6}, 4.38 \cdot 10^{-6}, 3.28 \cdot 10^{-6})$ for $T \in (50, 100, 250, 500, 1000, 1500, 2000)$ respectively.

**Figures 1c,S9.** WRN trained using momentum$= 0.9$, data augmentation and $\eta = 0.2$, evolved for 200 epochs. The $\text{AUTO}L_2$ algorithm is written explicitly in SM E and make measurements every 10 steps.

**Figure 2a,b,c.** FC trained using SGD $\frac{2}{\eta\lambda}$ epochs with learning rate and $L_2$ regularizations $\eta \in (0.0025, 0.01, 0.02, 0.025, 0.03, 0.05, 0.08, 0.15, 0.3, 0.5, 1, 1.5, 2, 5, 10, 25, 50)$, $\lambda \in (0, 10^{-5}, 0.0001, 0.0005, 0.001, 0.005, 0.01, 0.05, 0.1, 0.5, 1, 5, 10, 20, 50, 100)$. The $\lambda = 0$ model was evolved for $10^6/\eta$ epochs which is more than the smallest $\lambda$.

**Figure 2d,e,f.** WRN trained using SGD without data augmentation for $\frac{0.1}{\eta\lambda}$ epochs for the following hyperparameters $\eta \in (0.0125, 0.025, 0.05, 0.1, 0.2, 0.4, 0.8, 1.6), \lambda \in (0, 1.5625 \cdot 10^{-5}, 6.25 \cdot 10^{-5}, 1.25 \cdot 10^{-4}, 2.5 \cdot 10^{-4}, 5 \cdot 10^{-4}, 10^{-3}, 2 \cdot 10^{-3}, 4 \cdot 10^{-3}, 8 \cdot 10^{-3}, 0.016)$, as long as the total number of epochs was $\leq 4000$ epochs (except for $\eta = 0.2, \lambda = 6.25 \cdot 10^{-5}$ which was evolved for 8000 epochs). We evolved the $\lambda = 0$ models for 10000 epochs.

**Figure S7.** Fully connected depth 3 and width 64 trained on CIFAR-10 with batch size 512, $\eta = 0.1$ and cross-entropy loss.

**Figure S8.** ResNet-50 trained on ImageNet with batch size 8192, using the implementation in `https://github.com/tensorflow/tpu`.

**Figure 5** (a,b) plots $f_t$ in equation 3 with $k = 2$ (for 2-layer) and $k = 1$ (for linear), for different values of $\gamma$ and $\lambda = 0.01$. (c) The empirical kernel of a $2-$layer ReLU network of width 5,000 was evaluated on 200-samples of MNIST with even/odd labels. The linear, $2-$layer curves come from evolving equation 1 with the previous kernel and setting $k = 1, k = 2$, respectively . The experimental curve comes from training the 2-layer ReLU network with width $10^5$ and learning rate $\eta = 0.01$ (the time is step $\times \eta$).

**Figure 3a,b,c.** CNN without BN trained using SGD for $\frac{0.01}{\eta\lambda}$ epochs for the following hyperparameters $\eta = 0.01, \lambda \in (0, 5 \cdot 10^{-5}, 0.0001, 0.0005, 0.001, 0.01, 0.05, 0.1, 0.25, 0.5, 1, 2)$. with $\lambda = 0$ was evolved for 21000 epochs.

**Figure 3d,e,f.** CNN with BN trained using SGD for a time $\frac{0.01}{\eta\lambda}$ for the following hyperparameters $\eta = 0.01, \lambda = 0, 5 \cdot 10^{-5}, 0.0001, 0.0005, 0.001, 0.01, 0.05, 0.1, 0.25, 0.5, 1, 2$. The model with $\lambda = 0$ was evolved for 9500 epochs, which goes beyond where all the other $\lambda$'s have peaked.

**Figure S5.** FC trained using SGD and MSE loss for $\frac{1}{\eta\lambda}$ epochs and the following hyperparameters $\eta \in (0.001, 0.005, 0.01, 0.02, 0.035, 0.05, 0.15, 0.3), \lambda \in (10^{-5}, 0.0001, 0.0005, 0.001, 0.005, 0.01, 0.05, 0.1, 0.5, 1, 5, 50, 100)$. For $\lambda = 0$, it was trained for $10^5/\eta$ epochs.

**Rest of SM figures.** Small modifications of experiments in previous figures, specified explicitly in captions.

## B  Details of theoretical results

In this section we prove the main theoretical results. We begin with two technical lemmas that apply to $k$-homogeneous network functions, namely network functions $f_\theta(x)$ that obey the equation $f_{a\theta}(x) = a^k f_\theta(x)$ for any input $x$, parameter vector $\theta$, and $a > 0$.

**Lemma 1.** *Let $f_\theta(x)$ be a $k$-homogeneous network function. Then $\sum_\mu \theta_\mu \partial_\mu \partial_{\nu_1} \cdots \partial_{\nu_m} f(x) = (k - m)\partial_{\nu_1} \cdots \partial_{\nu_m} f(x)$.*

*Proof.* We prove by induction on $m$. For $m = 0$, we differentiate the homogeneity equation with respect to $a$.

$$0 = \frac{\partial}{\partial a}\bigg|_{a=1} \left( f_{a\theta}(x) - a^k f_\theta(x) \right) = \sum_\mu \frac{\partial f(x)}{\partial \theta_\mu}\theta_\mu - kf_\theta(x). \tag{S1}$$

For $m > 0$,

$$\sum_\mu \theta_\mu \partial_\mu \partial_{\nu_1} \cdots \partial_{\nu_m} f(x) = \partial_{\nu_m}\left( \sum_\mu \theta_\mu \partial_\mu \partial_{\nu_1} \cdots \partial_{\nu_{m-1}} f(x) \right) - \sum_\mu \left( \partial_{\nu_m}\theta_\mu \right)\partial_\mu \partial_{\nu_1} \cdots \partial_{\nu_{m-1}} f(x)$$

$$= \partial_{\nu_m}(k - m + 1)\partial_{\nu_1} \cdots \partial_{\nu_{m-1}} f(x) - \sum_\mu \delta_{\mu\nu_m}\partial_{\nu_m}\theta_\mu \partial_\mu \partial_{\nu_1} \cdots \partial_{\nu_{m-1}} f(x)$$

$$= (k - m)\partial_{\nu_1} \cdots \partial_{\nu_{m-1}}\partial_{\nu_m} f(x). $$

$$\tag{S2}$$

$\square$

**Lemma 2.** *Consider a k-homogeneous network function $f_\theta(x)$, and a correlation function $C(x_1, \ldots, x_m)$ that involves derivative tensors of $f_\theta$. Let $L = \sum_{x \in S} \ell(x) + \frac{1}{2}\lambda\|\theta\|_2^2$ be a loss function, where $S$ is the training set and $\ell$ is the sample loss. We train the network using gradient flow on this loss function, where the update rule is $\frac{d\theta^\mu}{dt} = -\frac{dL}{d\theta^\mu}$. If the conjecture of Dyer and Gur-Ari [2020] holds, and if the conjecture implies that $C = \mathcal{O}(n^{-1})$ where $n$ is the width, then $\frac{dC}{dt} = \mathcal{O}(n^{-1})$ as well.*

*Proof.* The cluster graph of $C$ has $m$ vertices; we denote by $n_e$ ($n_o$) the number of even (odd) components in the graph (we refer the reader to Dyer and Gur-Ari [2020] for a definition of the cluster graph and other terminology used in this proof). By assumption, $n_e + (n_o - m)/2 \leq -1$.

We can write the correlation function as $C(x_1, \ldots, x_m) = \sum_{\text{indices}} \mathbb{E}_\theta\left[\partial f(x_1) \cdots \partial f(x_m)\right]$, where $\mathbb{E}_\theta[\cdot]$ is a mean over initializations, $\partial f(x)$ is shorthand for a derivative tensor of the form $\partial_{\mu_1 \ldots \mu_a} f(x) := \frac{\partial^a f(x)}{\partial \theta^{\nu^1} \ldots \partial \theta^{\nu^a}}$ for some $a$, and the sum is over all the free indices of the derivative tensors. Then $\frac{dC}{dt} = \sum_{b=1}^m C_b$, where $C_b := \mathbb{E}_\theta\left[\partial f(x_1) \cdots \frac{d\partial f(x_b)}{dt} \cdots \partial f(x_m)\right]$. To bound the asymptotic behavior of $dC/dt$ it is therefore enough to bound the asymptotics of each $C_b$.

Notice that each $C_b$ is obtained from $C$ by replacing a derivative tensor $\partial f$ with $d(\partial f)/dt$ inside the expectation value. Let us see how this affects the cluster graph. For any derivative tensor $\partial_{\mu_1 \ldots \mu_a} f(x) := \partial^a f(x)/\partial \theta^{\nu^1} \cdots \partial \theta^{\nu^a}$, we have

$$\frac{d}{dt}\partial_{\mu_1 \ldots \mu_a} f(x) = \sum_\nu \partial_{\mu_1 \ldots \mu_a \nu} f(x) \frac{d\theta^\nu}{dt}$$

$$= -\sum_\nu \partial_{\mu_1 \ldots \mu_a \nu} f(x)\left[\sum_{x' \in S} \partial_\nu f(x')\ell'(x') + \lambda\theta^\nu\right]$$

$$= -\sum_{\nu, x'} \partial_{\mu_1 \ldots \mu_a \nu} f(x)\partial_\mu f(x')\ell'(x') - (k - a)\lambda\partial_{\mu_1 \ldots \mu_a} f(x) . \qquad \text{(S3)}$$

In the last step we used lemma 1. We now compute how replacing the derivative tensor $\partial f$ by each of the terms in the last line of (S3) affects the cluster graph, and specifically the combination $n_e + (n_o - m)/2$.

The second term is equal to the original derivative tensor up to an $n$-independent factor, and therefore does not change the asymptotic behavior. For the first term, the $\ell'$ factor leaves $n_e$ and $n_o - m$ invariant so it will not affect the asymptotic behavior. The additional $\partial_\mu f$ factor increases the number of vertices in the cluster graph by 1, namely it changes $m \mapsto m + 1$. In addition, it increases the size of the graph component of $\partial_{\mu_1 \ldots \mu_a} f(x)$ by 1, therefore either turning an even sized component into an odd sized one or vice versa. In terms of the number of components, it means we have $n_e \mapsto n_e \pm 1, n_o \mapsto n_o \mp 1$. Therefore, $n_e + (n_o - m)/2 \mapsto n_e + (n_o - m)/2 \pm 1 \mp \frac{1}{2} - \frac{1}{2} \leq n_e + (n_o - m)/2 \leq 1$. Therefore, it follows from the conjecture that $C_b = \mathcal{O}(n^{-1})$ for all $b$, and then $dC/dt = \mathcal{O}(n^{-1})$. $\qquad \square$

We now turn to the proof of Theorems 1 and 2.

*Proof (Theorem 1).* A straightforward calculation leads to the following gradient flow equations for the network function and kernel.

$$\frac{df_t(x)}{dt} = -\sum_{x' \in S} \Theta_t(x, x')\ell'(x') - k\lambda f_t(x) , \qquad \text{(S4)}$$

$$\frac{d\Theta_t(x, x')}{dt} = -2(k - 1)\lambda\Theta_t(x, x') + T_t(x, x') + T_t(x', x) , \qquad \text{(S5)}$$

$$T_t(x, x') = -\sum_{x'' \in S} \partial_{\mu\nu} f_t(x)\partial_\mu f_t(x')\partial_\nu f_t(x'')\ell'(x'') . \qquad \text{(S6)}$$

Here $\ell' = d\ell/df$. In deriving these we used the gradient flow update $\frac{d\theta^\mu}{dt} = -\frac{\partial L}{\partial \theta^\mu}$ and Lemma 1. It was shown in Dyer and Gur-Ari [2020] that $\mathbb{E}_\theta[T_0] = \mathcal{O}(n^{-1})$. If then follows from Lemma 2 that $\mathbb{E}_\theta\left[\frac{d^m T_0}{dt^m}\right] = \mathcal{O}(n^{-1})$ for all $m$, where the expectation value is taken at initialization. Furthermore,

the results of Dyer and Gur-Ari [2020] imply that $\text{Var}\left[\frac{d^m T_0}{dt^m}\right] = \mathcal{O}(n^{-2})$ and therefore $\frac{d^m T_0}{dt^m} \xrightarrow{p} 0$.[4]
In the strict infinite width limit we can therefore neglect the $T$ contribution in the following equation, and write

$$\frac{d^m \Theta_0(x, x')}{dt^m} = [-2(k-1)\lambda]^m \Theta_0(x, x'), \quad m = 0, 1, \dots .$$ (S7)

The solution of this set of equations (labelled by $m$) is the same as for the any-time equation $\frac{d}{dt}\Theta_t(x, x') = -2(k-1)\lambda\Theta_t(x, x')$, and the solution is given by

$$\Theta_t(x, x') = e^{-2(k-1)\lambda t}\Theta_0(x, x').$$ (S8)

$\square$

*Proof (Theorem 2).* The evolution of the kernel eigenvalues, and the fact that its eigenvectors do not evolve, follow immediately from (2). The solution (3) can be verified directly by plugging it into (1) after projecting the equation on the eigenvector $\hat{e}_a$. Finally, the fact that the function decays to zero at late times can be seen from (3) as follows. From the assumption $k \geq 2$, notice that $\exp\left[-\frac{\gamma_a(t')}{2(k-1)\lambda} - (k-2)\lambda t'\right] \leq 1$ when $t' \geq 0$. Therefore, we can bound each mode as follows.

$$|f_a(x; t)| \leq e^{-k\lambda t}\left[e^{\frac{(\gamma_a(t) - \gamma_a)}{2(k-1)\lambda}}|f_a(x; 0)| + |\gamma_a y|e^{\frac{\gamma_a(t)}{2(k-1)\lambda}}\int_0^t dt'\right].$$ (S9)

Therefore, $\lim_{t \to \infty} |f_a(x; t)| = 0$. $\square$

For completeness we now write down the solution (3) in functional form, for $x \in S$ in the training set.

$$f_t(x) = e^{-k\lambda t}\Bigg\{\sum_{x' \in S} \exp\left[\frac{\Theta_t - \Theta_0}{2(k-1)\lambda}\right](x, x')\, f_0(x')$$

$$+ \sum_{x', x'' \in S} \int_0^t dt'\, e^{-(k-2)\lambda t'} \exp\left[\frac{\Theta_t - \Theta_{t'}}{2(k-1)\lambda}\right](x, x')\, \Theta_0(x', x'')\, y(x'')\Bigg\}.$$

$$\Theta_t(x, x') = e^{-2(k-1)\lambda t}\Theta_0(x, x').$$ (S10)

Here, $\exp(\cdot)$ is a matrix exponential, and $\Theta_t$ is a matrix of size $N_{\text{samp}} \times N_{\text{samp}}$.

## B.1 Deep linear fixed point analysis

Let's consider a deep linear model $f(x) = \beta W_L....W_0.x$, with $\beta = n^{-L/2}$ for NTK normalization and $\beta = 1$ for standard normalization. The gradient descent equation will be:

$$\Delta W_{ab}^l = -\eta\lambda W_{ab}^l - \eta\beta \vec{W}_a^{l+1}\overleftarrow{W}_{b\alpha}^{l-1} \sum_{(x,y) \in S} \ell'(x, y)x_\alpha$$ (S11)

where we defined:
$$\vec{W}^l \equiv W^L...W^l, \overleftarrow{W}_\alpha^l \equiv W^l...W^0_\alpha$$ (S12)

Evolution will stop when the fixed point ($\Delta W = 0$) is reached:

$$W_{ab}^{L>l>0} = \vec{W}_a^{l+1}\hat{W}_b^{l-1}; W_a^L = \hat{W}_a^{L-1}; W_{a\alpha}^0 = \vec{W}_a^1\hat{W}_\alpha^{-1}$$ (S13)

$$\hat{W}_b^{l-1} \equiv -\frac{\beta}{\lambda}\overleftarrow{W}_{b\alpha}^{l-1} \sum_{(x,y) \in S} \ell'(x, y)x_\alpha; \overleftarrow{W}_{a\alpha}^{-1} = \delta_{a\alpha}$$ (S14)

Furthermore note that:

$$\vec{W}^l.\hat{W}^{l-1} = -\frac{1}{\lambda} \sum_{(x,y) \in S} \ell'(x, y)f(x) = \tilde{f}$$ (S15)

Now, we would like to show that, at the fixed point

$$\vec{W}_a^l = \tilde{f}^{L-l}\hat{W}_a^{l-1} \tag{S16}$$

This follows from induction:

$$\vec{W}^L = W^L = \hat{W}^{L-1} \tag{S17}$$

$$\vec{W}^l = \vec{W}^{l+1}(\vec{W}^{l+1}\hat{W}^{l-1}) = \tilde{f}^{L-l-1}\hat{W}^l(\vec{W}^{l+1}\hat{W}^{l-1}) = \tilde{f}^{L-l}\hat{W}^{l-1} \tag{S18}$$

Which has a trivial solution if $\tilde{f} = 0$. Let's assume that it is non-trivial. If we contract the previous equation with $\hat{W}^{l-1}$ we get:

$$\tilde{f} = \tilde{f}^{L-l}||\hat{W}^{l-1}||^2 \tag{S19}$$

We can finally set $l = 0$ and simplify:

$$\frac{\lambda^{L+1}}{\beta^2} = [-\sum_{(x,y)\in S} \ell'(x,y)f(x)]^{L-1} \sum_{(x,y),(x',y')\in S} \ell'(x,y)\ell'(x',y')x.x' \tag{S20}$$

At large $n$, to obtain a non-trivial fixed point $f(x)$ should be finite as $n \to \infty$. From the previous equation, this implies that $\frac{\lambda^{L+1}}{\beta^2} = \theta(n^0)$. In NTK normalization $\beta^2 = n^{-L}$, for $\lambda = \theta(n^{-\frac{L}{L+1}})$, we will get a non-trivial ($f(x) \neq 0$) fixed point. This also implies that these corrections will be important for $\lambda = \theta(n^0)$ in standard normalization since there $\beta = 1$. Note that if $\frac{\lambda^{L+1}}{\beta^2} \neq \Omega(n^0)$, we expect that we get the $\lambda = 0$ solution $\ell'(x,y) = 0$.

We can be very explicit if we consider $L = 1$ and one sample with $x, y = 1, 1$ for MSE loss. The fixed point has a logit:

$$\lambda\sqrt{n} = f - 1 \tag{S21}$$

which is only different from $0, 1$ for fixed $\lambda^2 n$.

# C   More on experiments

## C.1   Training accuracy = 1 scale

We can see how the time it takes to rech training accuracy 1 depends very mildly on $\lambda$, and for small enough learning rates it scales like $1/\eta$.

Figure S1: Training accuracy vs learning rate the setup of figure 2. The specific values for the $\eta, \lambda$ sweeps are in A.

## C.2   More WRN experiments

We can also study the previous in the presence of momentum and data augmentation. These are the experiments that we used in figure 4, evolved until convergence. As discussed before, in the presence of momentum the $t_*$ depends on $\eta$, so we will fixed the learning rate $\eta = 0.2$.

Figure S2: WRN 28-10 with momentum and data augmentation trained with a fixed learning rate.

## C.3   More on optimal $L_2$

Here we give more details about the optimal $L_2$ prediction of section 3. Figure S3 illustrates how performance changes as a function of $\lambda$ for different time budgets with the predicted $\lambda$ marked with a dashed line. If one wanted to be more precise, from figure 2 we see that while the scaling works across $\lambda$'s, generally lower $\lambda$'s have a scaling $\sim 2$ times higher than the larger $\lambda$'s. One could try to get a more precise prediction by multiplying $c$ by two, $c_{\text{small}\lambda} \sim 2c_{\text{large}\lambda}$, see figure S4. We reserve a more detailed analysis of this more fine-grained prescription for the future.

Figure S3: WRN trained with momentum and data augmentation. Given a number of epochs, we compare the maximum test accuracy as a function of $L_2$ and compare it with the smallest $L_2$ with the predicted one. We see that this gives us the optimal $\lambda$ within an order of magnitude.

Figure S4: Same as previous figure with $c = 2c_{\text{large}\lambda} = 0.0132$.

## C.4  MSE and the catapult effect

In Lewkowycz et al. [2020] it was argued that, in the absence of $L_2$ when training a network with SGD and MSE loss, high learning rates have a rather different final accuracy, due to the fact that at early times they undergo the "catapult effect". However, this seems to contradict with our story around 1.1 where we argue that performance doesn't depend strongly on $\eta$. In figure S5, we can see how, while when stopped at training accuracy 1, performance depends strongly on the learning rate, this is no longer the case in the presence of $L_2$ if we evolve it for $t_{test}$. We also show how the training MSE loss has a minimum after which it increases.

Figure S5: MSE and catapult effect: we see how even if there is a strong dependence of the test accuracy on the learning rate when the training accuracy is 1, this dependence flattens out when evolved until convergence in the presence of $\lambda$ . The specific values for the $\eta, \lambda$ sweeps are in A.

## C.5 Dynamics of loss and accuracy

In figure S6 we illustrate the training curves of the experiments we have discussed in the main text and SM.

Figure S6: This shows the dynamics of experiments in figures 2, 3.The learning rates are 0.01 for the CNN and 0.2 for the WRN and 0.15 for the FC. Each legend has the min, max and median value of the logspace $\lambda$'s with its respective color.

# D  Examples of setups where the scalings don't work

We will consider a couple of setups which don't exhibit the behaviour described in the main text: a 3 hidden layer, width 64 fully-connected network trained on CIFAR-10 and ResNet-50 trained on ImageNet. We attribute this difference to deviations from the overparametrized/large width regime. In this situation, the optimal test accuracy with respect to $\lambda$ has a maximum at some $\lambda_{\mathrm{opt}} \neq 0$.

For the FC experiment, the time it takes to reach this maximum accuracy scales like $1/\lambda$ for $\lambda \gtrsim \lambda_{opt}$, but becomes constant (equal to the value for $\lambda = 0$) for $\lambda \lesssim \lambda_{\mathrm{opt}}$. This peak of the maximum test accuracy happens before the training accuracy reaches 1. Generically, we don't observe that a network trained with cross-entropy and without regularization to have a peak in the test accuracy at a finite time.

We do not have as clear an understanding of the ImageNet experimental results because they involve a learning rate schedule. Performance for small $\lambda$s does not improve even if when evolving for a longer time. However, we do observe that performance is roughly constant when $\eta \cdot \lambda$ is held fixed.

Figure S7: Experiments with FC of width 64 and depth 3 trained on CIFAR-10.

(a)

| | 0.0125 | 0.025 | 0.05 | 0.1 | 0.2 | 0.4 | 0.8 |
|---|---|---|---|---|---|---|---|
| .25e-05 | .689 | .697 | .707 | .72? | .736 | .747 | .750 |
| 2.5e-05 | .711 | .720 | .730 | .744 | .751 | .755 | .753 |
| 5e-05 | .72? | .731 | .743 | .754 | .759 | .759 | .752 |
| 0.0001 | .728 | .740 | .752 | .760 | .762 | .755 | .734 |
| 0.0002 | .736 | .748 | .757 | .759 | .751 | .727 | .699 |
| 0.0004 | .742 | .751 | .754 | .752 | .735 | .707 | .676 |
| 0.0008 | .743 | .750 | .746 | .730 | .706 | .672 | |

learning rate

(b)

| | 1779 | 3558 | 7116 | 14232 | 28464 | 56928 | 113856 |
|---|---|---|---|---|---|---|---|
| .25e-05 | | | | | | | |
| 2.5e-05 | .662 | .665 | .684 | .692 | .694 | .674 | .665 |
| 5e-05 | .715 | .726 | .734 | .741 | .740 | .731 | .705 |
| 0.0001 | .729 | .742 | .752 | .759 | .760 | .755 | .733 |
| 0.0002 | .730 | .740 | .753 | .761 | .765 | .763 | .748 |
| 0.0004 | .727 | .740 | .749 | .755 | .757 | .760 | .742 |
| 0.0008 | .723 | .734 | .745 | .746 | | | |

steps

Figure S8: ResNet-50 trained on ImageNet. (a) Evolved for 14,232 epochs for different $\eta, \lambda$. While changing $\eta$ or $\lambda$ independently has a strong effect of performance, we see that performance is rather similar along the diagonal. (b) Fixed $\eta = 0.1$ and evolve for different $\lambda$ and number of epochs $T$ (rescaling the learning rate schedule to $T$). In contrast to the overparameterized case, we see that one cannot reach the same performance with a smaller $\lambda$ by increasing $T$.

# E  More on AUTO$L_2$

The algorithm is the following:

---

**Algorithm 1:** AUTOL2

---
MINLOSS,MINERROR=$\infty, \infty$
MIN_STEP,L2=0,0.1
**for** *t in steps* **do**
   UPDATE_WEIGHTS;
   **if** $t\ mod\ k = 0$ **then**
      MAKE_MEASUREMENTS;
      **if** ERROR_OR_LOSS_INCREASES AND $t >$ MIN_STEP **then**
         L2=L2/10;
         MIN_STEP=$0.1/$L2$+ t$;
      **else**
         MINLOSS,MINERROR=min(LOSS$_{t-k}$,MINLOSS),min(ERROR$_{t-k}$,MINERROR);

**Function** ERROR_OR_LOSS_INCREASES (LOSS,ERROR,MINLOSS,MINERROR):
   **if** LOSS$_t$>MINLOSS AND LOSS$_{t-k}$>MINLOSS **then**
      **return** True;
   **if** ERROR$_t$>ERROR AND ERROR$_{t-k}$>ERROR **then**
      **return** True;
   **return** False;

---

We require the loss/error to be bigger than its minimum value two measurements in a row (we make measurements every 5 steps), we do this to make sure that this increase is not due to a fluctuation. After decaying, we force $\lambda$ to stay constant for a time $0.1/\lambda$ steps, we choose the refractory period to scale with $1/\lambda$ because this is the physical scale of the system.

To complement the AUTO$L_2$ discussion of section 3 we have done another experiment where the learning rate is decayed using the schedule described in 2. Here we see how while AUTO$L_2$ trains faster in the beginning, the optimal $\lambda = 0.0005$ outperforms it. We have not hyperparameter tuned the possible parameters of AUTO$L_2$.

Figure S9: Here we have a WRN trained with momentum and data augmentation for 200 epochs. We compare the AUTO$L_2$ with different fixed $L_2$ parameters and we see how it trains faster and gets better accuracy.

We have also applied AUTO$L_2$ to other setups, and in the absence of learning rate schedules beats the optimal $L_2$ parameter. See figure S11.

Figure S10: Setup of S9 in the presence of a learning rate schedule. While AUTO$L_2$ trains faster and better in the beginning, it can't keep pace with the big jumps of constant $\lambda$.

Figure S11: AUTO$L_2$ for the other architectures considered in the main text compared with the optimal $L_2$ parameters.

# F Different initialization scales

We have discussed the dependence on the time to convergence in $\eta, \lambda$. Another quantity which is relevant for learning is $\sigma_w$ the scale of initialization, which for WRN we set to 1. We can repeat the WRN experiment of figure 1a for different $\sigma_w$. We see that the final performance depends very mildly on $\sigma_w$. This is what we expect when reaching equilibrium: the dependence of properties at initialization is eventually washed away.

Figure S12: While models with different $\sigma_w$ behave rather differently at early times, at late times, this $\sigma_w$ dependence washes off for longer times.

## Footnotes

[4]See appendix D in Dyer and Gur-Ari [2020].