[Reviews · NeurIPS 2020]

Review 1

Summary and Contributions: This paper studies the role of weight decay (\lambda) in affecting the model performance. The authors show that (1) the number of SGD steps to achieve the maximum performance is proportional to 1/\lambda; (2) when the number of SGD steps is fixed, the performance peaks at a certain value of weight decay; and (3) for fixed number of training steps, the model performance increases as the weight decay decreases. Based on these observations, the authors propose a simple way to estimate the optimal weight decay coefficient and AUTOL2, a method to dynamically adjust the weight decay during training. Finally, the authors provide a theoretical understanding with infinitely wide neural networks, showing that the performance peaks at a time of order 1/\lambda and deteriorates after that.

Strengths: - The paper is well-written and easy to follow. This paper also presents both carefully designed experiments and sounded theoretical results for justifying the findings and arguments. - This paper proposes a useful approach AUTOL2 for dynamically tuning weight decay, and demonstrates improved performance on image classification tasks. - The explanation that the smallest eigenvalue of the Hessian is approximately \lambda is interesting. This might explain why weight decay can also accelerate the convergence performance in addition to its regularization effect.

Weaknesses: - Theorem 2 is not very interesting. Because the effect of weight decay will be diminished as the learning rate decreases, and common implementation of weight decay [1] will usually multiply the amount of weight decay by the learning rate. Therefore, I believe theorem 2 is not very useful in practice and the results that lim_t->\infty f_t(x) = 0 is not surprising in your setting. - How do different learning rate schedules affect the conclusion? Will the conclusion at L33 and L36 still hold? - It would be great if the authors can provide more experiments on demonstrating the effectiveness of AUTOL2, as this will help to convince people to use it in training neural networks. Overall, I believe this is an interesting paper and the findings in the paper will be interested to the community. [1] https://pytorch.org/docs/stable/_modules/torch/optim/sgd.html#SGD ===================== After Rebuttal ====================== Thanks for the authors' response, I keep my score unchanged.

Correctness: Yes

Clarity: Yes

Relation to Prior Work: Yes

Reproducibility: Yes

Additional Feedback: See Weaknesses.


Review 2

Summary and Contributions: ----update---- Apologies to the authors for my misunderstanding and missing several crucial points for validating the practicality of the proposed work. Hopefully, the authors could clarify the learning rate decay schedule in their experiments, so that the results can be reproduced by others, and it also helps the paper reach a wide audience. ----end of the update---- The paper studied the effect of l2 regularisation in overparametrised models, specifically deep neural networks, and reached an interesting finding that the number of epochs for a model to achieve the optimal accuracy on the test set is inversely proportional to the optimal strength of l2 regularisation, which indicates that smaller l2 regularised models require more epochs to reach optimal on the test set.

Strengths: ((1)) The submission proposed a simple heuristic for the relationship between the number of epochs a model needs to be trained to achieve optimal accuracy on the test set and the strength of l2 regularisation. ((2)) It proposed an algorithm for finding the optimal strength by training a model with relatively large strength and then retrain the model with the optimal one given the limited number of training epochs. ((3)) Also, it proposed an AutoL2 method for automatically decaying the strength of l2 regularisation during learning which has been shown to be better than the model trained with optimally tuned one.

Weaknesses: ((1)) If I could have access to the test set, then why bother tuning l2 regularisation to get optimal on the test set? Technically, I could run a brute-force algorithm to find an optimal set of parameters without tuning any other hyperparameters. I do think the submission violates the ethics of machine learning research. I understand that theoretical work generally considers the generalisation gap between the training set and the test set, however, the submission is an empirical work on hyperparameter tuning for optimal l2 regularisation that gives the highest test set accuracy. Therefore, a validation set is required for tuning, and then it should be tested on the test set afterward. Another valid approach is to report the accuracy on the test set after a fixed number of epochs, but I don't think it applies here as the paper is trying to find the relationship between the number of epochs one could use to train a model and the strength of l2 regularisation. ((2)) I have concerns on comparing AutoL2 and lambda=0.0001. Other experiments used learning rate decay to train neural networks, which is standard. In Fig. 1c, the test accuracy smoothly increases for a model trained with lambda = 0.0001, which is not common when learning rate decay is applied as, when it reduces by a factor of 5, there usually is a jump in the test accuracy. So I was wondering if the comparison is fair here as the learning rate decay is commonly used is training deep learning models, especially for classification models. ((3)) The practically of the proposed work in this submission is rather limited. The whole training process involves many hyperparameters, and the submission empirically tested the effect of l2 regularisation under rather limited combinations of other hyperparameters, which raises my doubt on the generalisation of the claim. The most interesting plots might be Test Acc vs Lambda in Fig. 2c and 2f, but the authors didn't report multiple runs of the same configuration so that the significance of the difference can be shown in those plots.

Correctness: Yes, I think the claims are correct under the conditions described in the paper.

Clarity: Yes.

Relation to Prior Work: The paper has a clear presentation in relating itself to both theoretical and empirical work.

Reproducibility: No

Additional Feedback:


Review 3

Summary and Contributions: This paper studies training overparameterized networks with L2 regularization. The analysis how the regularization hyperparameter \lambda affects the test accuracy with respect to the number of training epochs. Furthermore, authors propose two ways of determining optimal regularization \lambda: i) by determining the c constant by using large \lambda=0.1, and ii) with a regularization \lambda scheduler (similar to a learning rate scheduler). And the empirical observations are supported by theoretical results on an infinitely wide network with positive homogenous activations. ---After author response---- Thanks for the author response.

Strengths: The observation of the c/(n*\lambda) is very interesting, which also matches with my network hyperparameter search past experiences. Furthermore, the applications of optimal regularization \lambda (c coefficient, and AutoL_{2}) seem very simple and good replacements for setting regularization lambda by hyperparameter search. The theoretical results support the empirical observations. Experimental setup is clearly explained and code for some part of the empirical results (Figure 1 and Figure 2) is shared along with supplementary sections.

Weaknesses: Not really a weakness: but more like future work, more insights on the relation between learning rate scheduler and AutoL_{2} regularization \lambda scheduler would be very interesting. Is there any relation with the need of suspension episodes after updating the regularization hyperparameter lambda as a part of AutoL_2?

Correctness: The observations are done using MLP, Wide ResNet and CNN (with/without batch normalization) on the CIFAR-10 dataset using stochastic gradient descent without momentum. In addition, the applications are done in a more realistic setting with data augmentation, and stochastic gradient descent with momentum.

Clarity: I quite enjoy the writing style that the explanations are detailed enough that immediately answers the questions that appear while reading without requiring to search this information throughout the paper.

Relation to Prior Work: The paper is framed well compared to earlier work on L2 regularization. That would be nice to extend to major related works on (other ways of) regularization.

Reproducibility: Yes

Additional Feedback: During AutoL_{2} scheduled regularization, after the lambda update refractory period is not detailed (Line 153): how long is it?, can this heuristic generalize across network architectures/datasets? The Figure 1c contains test accuracy curves of X and X. It would be interesting to see on the same graph, training with learning rate scheduler \lambda=0 and training with learning rate scheduler with optimum \lambda=X. In Figure 1a and 1b, how is the best test accuracy determined? Through validation set loss or directly comparing the test accuracy results? The same question for Figure 2 and 3.


Review 4

Summary and Contributions: The paper shows a relationship of the l2-regularization of over-parameterized networks with the optimization time and other training parameters. In particular they show that: * with finite training budget with l2 norm the training loss peaks a given point in time during training. Based on this an optimal regularization parameter can be derived. * that this point is in time is proportional to the regularization strengh. * this can be theoretically justified and that these claims also hold in practice. ---- read the reviews and the rebuttal and want to keep my score ----

Strengths: * Great to read. * Theory and practice go hand in hand and the experimental results fit surprisingly well the theoretical estimations. * IMHO gives a good, additional understanding on DNN.

Weaknesses: * It would be great to have more of the discussion in appendix D (when the shown properties do not hold) in the main paper as it very relevant to the reader. * Only show theoretically the impact of the network depth on the stated properties. It would be interesting to see how this manifests in practice, exp for "shallower" networks.

Correctness: To the best of my knowledge everything seems fine. Said that I am not expert on the theoretical part and don't have the expertise to judge that part (but everything sounds reasonable). The experiments are described in detail in the appendix and code is given, which is an indicator of correctness for me.

Clarity: It's a pleasure to read. The paper is well structured and easy to read.

Relation to Prior Work: I am not an expert in the field, but the from reading it seems like the paper gives a good comparison to prior work.

Reproducibility: Yes

Additional Feedback: * I would name section 1.1 section 2 as it fairly long and no intro. * Default test accuracy of WRN of 0.92 seems a low to me. In the paper 0.96 is reached. Do you have any explanation for that? * It would be great to have a better description on the relationship of learning rate schedules and the regularization. There are several places where this is mentioned. but it is not clear to me if it has impact in the end or not. * An imagenet training with an overparameterized (resnet50) would boost the practical, impact of the paper even more. Would be happy to see more on that.

[Author Response · NeurIPS 2020]

We thank the reviewers for their detailed comments. Please see our response below.

**R1:** ○ *"... common implementation of weight decay [1] will usually multiply the amount of weight decay by the learning*
*rate."* The same holds in our setup: We have an $L_2$ regularization term in the loss. In the gradient descent update
equations, this results in a weight decay term where the $L_2$ coefficient is multiplied by the learning rate, as in [1].
Perhaps this is not immediately obvious in Theorem 2 where we consider gradient flow rather than gradient descent, but
the statement holds in that case as well.

○ *"How do different learning rate schedules affect the conclusion?"*: We address LR schedule questions below.

○ *"It would be great if the authors can provide more experiments on ... AUTOL2"* We ran additional experiments
training Wide ResNets and ConvNets on CIFAR-100 and SVHN, with similar conclusions: AutoL2 either beats or
matches the performance and training speed of a tuned, constant $L_2$ parameter. As an example, for CIFAR-100 with a
fixed lr and evolved for 500 epochs, the optimal $L_2$ parameter gives a test accuracy of 0.76 while AutoL2 gives 0.79.

**R2:** ○ *"((1)) If I could have access to the test set..."*. We reject the claim that our submission "violates the ethics of
machine learning research". The method we propose for finding the optimal $L_2$ parameter does not rely on a validation
or test set — it can be implemented using training data alone! Our baseline is to compare against tuning on the test set,
as is commonly done in the deep learning literature. If our setup introduces any bias (compared with having a separate
validation set), it is in favor of the baseline.

○ *"((2)) I have concerns on comparing AutoL2..."*. Indeed, the experiment of Fig. 1c does not include lr decay (as
discussed in the text). Experiments with lr decay and AutoL2 are presented in the SM. Please see below for additional
discussion.

○ *"((3))) The practically of the proposed work... The most interesting plots might be Test Acc vs Lambda in Fig. 2c and*
*2f, but the authors didn't report multiple runs of the same configuration..."* Please see Fig. 1a, 2b, and 2e for many
additional runs illustrating the same effect. We made an effort in the paper to understand the settings under which our
conclusions hold; these are summarized in the Discussion and in SM.D.

**R3:** ○ *"... more insights on the relation between learning rate scheduler and AutoL2..."* We address this point in the
learning decay discussion at the bottom of the page.

○ *"... the lambda update refractory period is not detailed ..."* The refractory period lasts for $\frac{\lambda(t=0)}{\lambda(t)}$ steps. This is
explained in the SM, and we will clarify it in the main text. The results are not sensitive to the choice of $\lambda(t=0)$ and
we pick $0.1$ in our experiments.

○ *"It would be interesting to see on the same graph, training with learning rate scheduler..."* In the SM we have the
training curves for the models trained with a learning rate schedule (see figure S10).

○ *"In Figure 1a and 1b, how is the best test accuracy determined?..."* In Figs. 1a,2,3, the model is trained for a specified
number of epochs and we report the best test accuracy during training. In Fig. 1b, we have multiple runs with different
$L_2$ and the 'optimal $L_2$' corresponds to the value which achieved the best test accuracy during training.

**R4:** ○ *"Default test accuracy of WRN of 0.92 seems a low ..."* The accuracies of Figs. 1a,1b exceed 0.96 for some
choices of the $L_2$ parameter. For figure 1c, we limited the training time to 200 epochs and used a fixed lr, which explains
why the accuracy is lower.

○ *"It would be great to have a better description on the relationship of learning rate schedules..."* Please see LR
schedule comments below.

○ *"An imagenet training with an overparameterized (resnet50) would boost..."* We agree. We have run several additional
experiments on CIFAR-100 and SVHN and the results show consistent improvement when using AutoL2. It is not
clear whether ResNet50 on ImageNet (when trained with data augmentation) is sufficiently overparameterized for our
purposes — we comment on this in SM.D.

**Comments on learning rate schedules.** Here we address reviewers' questions regarding learning rate schedules, and
we will revise the paper to include these points. Our empirical observations apply in the presence of learning rate
schedules, as illustrated in the paper. Figure 1a shows that the test accuracy remains roughly the same if we scale the
training time as $1/\lambda$. As to our proposed algorithms, we found that our predicted optimal $L_2$ value is within an order
of magnitude of the tuned value when using learning rate schedules, when accounting for the schedule by weighing
each step according to the learning rate. Therefore, as in the case of fixed learning rate, our prediction is beneficial
for hyperparameter tuning. For the AutoL2 scheduler, in the presence of learning rate decay we find that it performs
similarly to a tuned, fixed $L_2$ parameter but does not exceed it. This is still beneficial, in that it saves the need to tune
the $L_2$ parameter.

[Meta-Review · NeurIPS 2020]

This is a very interesting submission studying the effect of L2 regularization on training overparameterized networks. Significant and intuitive theoretical contributions are made that are nicely backed by empirical evaluation. We strongly urge the authors to incorporate their rebuttal points on learning rate schedules.